# Assessing the extent of drug interactions among patients with multimorbidity in primary and secondary care in the West Midlands (UK): a study protocol for the Mixed Methods Multimorbidity Study (MiMMS)

Ruth Backman,[1] Philip Weber,[2] Alice M Turner,[1,3] Mark Lee,[2] Ian Litchfield[1]

[1]Institute of Applied Health Research, University of Birmingham, Birmingham, UK
[2]School of Computer Science, University of Birmingham, Birmingham, UK
[3]Heartlands Hospital, Heart of England NHS Foundation Trust, Birmingham, UK

**Correspondence to**
Dr Ian Litchfield;
i.litchfield@bham.ac.uk

## ABSTRACT

**Introduction** The numbers of patients with three or more chronic conditions (multimorbidity) are increasing, and will rise to 2.9 million by 2018 in the UK alone. Currently in the UK, conditions are mainly managed using over 250 sets of single-condition guidance, which has the potential to generate conflicting recommendations for lifestyle and concurrent medication for individual patients with more than one condition. To address some of these issues, we are developing a new computer-based tool to help manage these patients more effectively. For this tool to be applicable and relevant to current practice, we must first better understand how existing patients with multimorbidity are being managed, particularly relating to concerns over prescribing and potential polypharmacy.

**Methods and analysis** Up to four secondary care centres, two community pharmacies and between four and eight primary care centres in the West Midlands will be recruited. Interviewees will be purposively sampled from these sites, up to a maximum of 30. In this mixed methods study, we will perform a dual framework analysis on the qualitative data; the first analysis will use the Theoretical Domains Framework to assess barriers and enablers for healthcare professionals around the management of multimorbid patients; the second analysis will use Normalisation Process Theory to understand how interventions are currently being successfully implemented in both settings. We will also extract quantitative anonymised patient data from primary care to determine the extent of polypharmacy currently present for patients with multimorbidity in the West Midlands.

**Discussion** We aim to combine these data so that we can build a useful, fully implementable tool which addresses the barriers most amenable to change within both primary and secondary care contexts.

**Ethics and dissemination** Favourable ethical approval has been granted by The University of Birmingham Research Ethics Committee (ERN_16–0074) on 17 May 2016. Our work will be disseminated through peer-reviewed literature, trade journals and conferences. We will also use the dedicated web page hosted by the University to serve as a central point of contact and as a repository

---

### Strengths and limitations of this study

► We are using a series of semi-structured interviews to elicit current barriers found within both primary and secondary care when managing patients with multimorbidity.
► This will be complemented by the first quantitative exploration of polypharmacy in multimorbid patients in primary care by interrogating routinely collected prescribing data within the same practices.
► This pilot study is based in the West Midlands whose diverse population means that a range of practices and clinicians will be participating.

---

of our findings. We aim to produce a minimum of three articles from this work to contribute to the international scientific literature.

**Protocol registration number** NIHR Clinical Research Network Portfolio Registration CPMS ID 30613.

## INTRODUCTION

In the UK there is a focus on the formation and implementation of guidelines to help standardise and improve the quality of care.[1–3] Providers may choose to translate this guidance into local policy to help implementation, for example, a multidisciplinary care plan which details essential steps in the management of patients with a specific clinical problem.[4] These pathways frequently use graphical descriptions of evidence and options and are typically represented in single, or series of, flow charts[5 6] and have been shown to have the possibility to improve care if they are implemented correctly.[7] These systems will contain prompts or reminders which in isolation can

help to improve practice[8–10] but in combination can lead to alert fatigue.[11]

Of the 250 clinical guidelines published by the National Institute for Health and Care Excellence,[12] only four account for more than one condition. This presents issues for clinicians attempting to use this guidance to manage multimorbidity. Defined as two or more chronic conditions in the same individual,[13] patients with multimorbidity are becoming more prevalent and it is estimated that by 2018 the number of people in the UK with three or more long-term conditions will have grown to 2.9 million.[14] Using multiple single condition guidelines means patients can be left with a significant treatment burden[15 16] and conflicting advice around lifestyle factors.[17] There are also implications for clinicians who struggle to balance the requirements of several guidelines for multimorbid patients, hindered by the lack of outcome data.[18]

One significant consequence of following multiple guidelines is the concurrent prescribing of multiple medications. This phenomenon has been termed 'polypharmacy'[19] and has been identified as a complicating issue for patients with multimorbidity.[20] The chances of polypharmacy greatly increase when several sets of guidance are being implemented in patients with multimorbidity[21] and healthcare professionals recognise that fulfilling multiple guidelines might compromise patient-centred care.[22] There is evidence this simultaneous medication administration can lead to an increased risk of patient hospitalisation[23 24] and work has begun to reduce this risk,[25] focusing on reducing inappropriate prescribing of non-steroidal anti-inflammatory drugs, proton pump inhibitors and duplicate therapy.[26] However progress is inconsistent and large interpractice variation in levels of polypharmacy remains.[27]

In response to the issues raised in managing increasing numbers of multimorbid patients there have been calls in the UK for improved integration of existing guidelines[28] with the maxim 'treat the patient, not the disease'. In response the 'MITCON' (Automated Conflict Resolution in Clinical Pathways) project has begun which involves the development of a new computer based tool which will automatically detect conflict between guidelines (Litchfield I, Turner A, Backman R *et al*. Automated conflict resolution between multiple clinical pathways. 2017. Unpublished). To ensure that this tool is applicable and relevant we need to understand how multiple guidelines are being translated into existing systems and processes and the extent and implications for patients and care providers of prescribing multiple medications. This protocol details the Mixed Methods Multimorbidity Study (MiMMS) which seeks to both identify barriers and facilitators to successful guideline implementation, and determine current levels of polypharmacy in multimorbid patients in the West Midlands.

## Research questions

There are two distinct yet related research questions that we will answer. The first concerns the way in which guidelines are used in managing multimorbid patients in two settings—primary and secondary care. The second will involve a quantitative assessment of polypharmacy in multimorbid patients in primary care. Although previous work has explored multimorbidity, ours is the first to look at both primary and secondary care settings and corroborate qualitative work with quantitative data on polypharmacy at the same general practitioner (GP) practices.

## Managing multimorbidity

We will use the Theoretical Domains Framework (TDF)[29 30] to identify barriers around managing patients with multimorbidity. This framework has been extensively validated within this setting,[31–36] and allows linkage of behaviour change techniques[30 37] if required later in the project. We will also use Normalisation Process Theory (NPT)[38] to identify the current successful implementation strategies for guidelines and associated computer based tools used within current practice. This theory has also been extensively validated and can assess the process whereby new items become routine practice. To the best of our knowledge, this work is the first to use a dual framework approach for analysis. Furthermore, this work will seek to identify barriers in multiple National Health Service (NHS) settings so that the full patient pathway with a number of healthcare professionals can be explored.

## Prevalence of multimorbidity

We aim to assess current levels of polypharmacy for up to six chronic conditions within primary care retrospectively over a 24-month period. In doing so, we will determine the percentage of patients that experience polypharmacy and whether there are patients with certain combinations of conditions that are most at risk. The literature indicates that our target conditions will occur in at least a pairwise combination. We will therefore perform pairwise comparisons of conditions to assess numbers of patients who have interactions within their prescribed medications.[39–53] This will allow us to determine which combinations of conditions are most frequently associated with polypharmacy. For this work we are defining multimorbidity as having two or more of our target conditions diagnosed with an active Read Code. We are defining polypharmacy as taking a minimum of two drugs at any one time point so that we can assess the interaction.

## METHODS AND ANALYSIS
### Settings and participants
#### Site recruitment

We will assess all primary care centres in the West Midlands for study eligibility (defined as using Egton Medical Information Systems (EMIS) as their clinical system). We will also purposively sample up to four secondary care centres with differing patient demographics in the West Midlands. Finally, we will purposively sample up to two pharmacy centres.

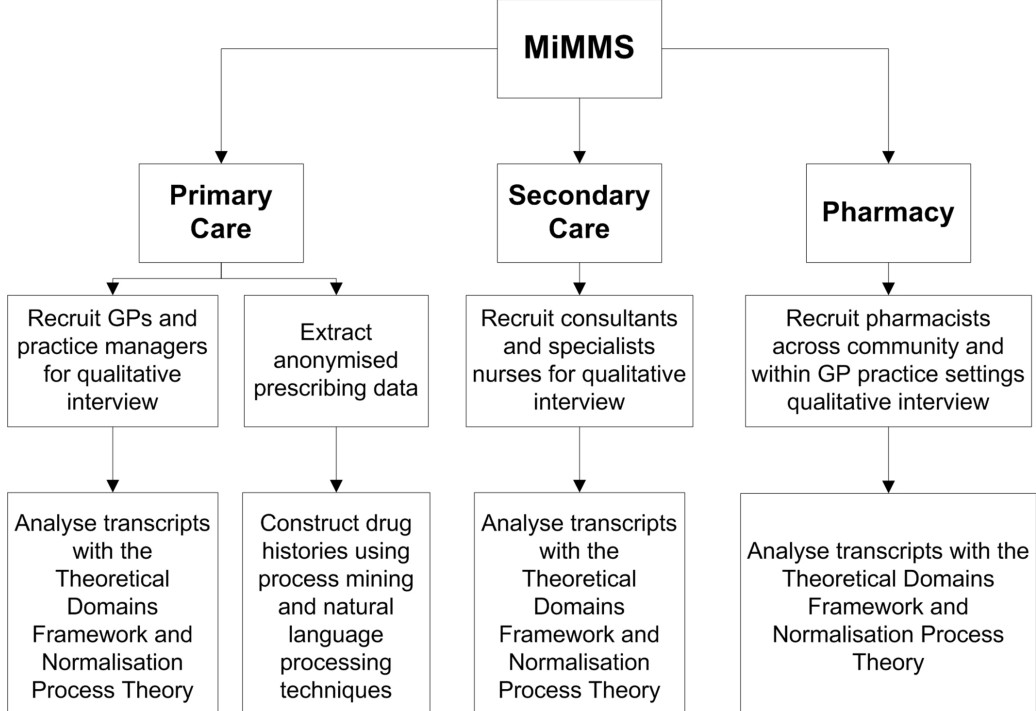

**Figure 1** Flow diagram of the Mixed Methods Multimorbidity Study (MiMMS). Summary of participation from each of the National Health Service sectors with an analysis overview from the quantitative and qualitative methodology. GP, general practitioner.

## Research design

We are using a mixed methods approach in this work and will be conducting the study in two interrelated phases (figure 1). Phase I is the qualitative exploration of the implementation of current guidelines including the barriers and enablers to the management of multimorbid patients in primary and secondary care settings including the use of clinical software systems. Phase II will use quantitative methods to explore the correlation between clinical condition combinations and the numbers of interacting medications prescribed in primary care.

## Phase I

### Site recruitment

In accordance with the theory of maximum variation,[54] between four and eight primary care sites and up to four secondary care sites will be recruited for this pilot from the West Midlands region. This will also help mitigate the risk of responder bias.[55] Sites will be initiated once all research and development (R&D) approvals have been met, and all necessary documents have been returned to the coordinating centre.

Recruitment for these sites will occur through written expressions of interest via the Primary Care Research Network as well as opportunistic recruitment of GPs during appropriate educational meetings. We will use purposive sampling for the secondary care and community pharmacy sites and will aim to recruit within different working environments, for example, community pharmacy and pharmacists working within

a GP practice, where possible. We will also interview healthcare professionals within different departments in secondary care.

Primary care sites that have EMIS web as their clinical system can participate in both phases of the project. We will aim to recruit healthcare professionals who have used other clinical systems such as The Phoenix Partnership (TPP) (SystmOne) or Vision to identify specific barriers relating to these systems.

### Healthcare professionals' recruitment

All healthcare professionals with experience either of managing patients with multimorbidity, or who have implemented new guidelines locally, will be invited to participate in phase I. In primary care we expect to recruit fully qualified and registrar-level GPs, advanced nurse practitioners and practice managers. Within secondary care we will recruit participants from specialties relevant to the six diseases targeted in the quantitative work as well as elderly care physicians if possible. These will include, but not be limited to, consultants, senior registrars (at least ST4 and above) and advanced nurse practitioners. Additionally, we will aim to recruit a minimum of two community pharmacists. All sampling will be performed using an iterative snowballing approach.[56]

In each case we will aim to recruit at least two individuals from each of the staff groups stated above, and we will continue with purposive sampling until data saturation has been achieved.[57]

## Data management

In-depth semi-structured interviews will be conducted using a topic guide tailored to both the NPT and TDF (supplementary file 1) so that both barriers around management of multimorbid patients and current guideline usage can be elicited. Prompts and examples will be used to ensure that data are captured and we will perform constant comparison analysis[58] during the interview process. In-depth semi-structured interviews will allow these specific topics to be covered, while retaining an element of narrative so that healthcare professionals can share the factors that are important to them. Interviewees will always be given the opportunity to add anything they wish at the start and end of the interview. Field notes will be taken by the researcher, both to allow for continued professional development and also to take note of surroundings and reflection of important findings.

Where possible, participants will be sent a copy of the consent form and information sheet at least 24 hours before the interview so that they have time to consider taking part and also time to reflect on the challenges of managing these types of patients. Using a snowballing technique,[56] sometimes known as the chain referral technique where an interviewee is asked to suggest a member of their team who may have further insights on a particular topic, this may not always be possible, so if participants require less time to decide to take part they may do so.

Fully informed consent will be taken by a trained researcher who holds a current good clinical practice certificate and all associated approvals. We recognise that healthcare professionals may feel uncomfortable sharing cases where they feel that patients may not have been managed to the published guidance, however all participants will be reassured that not all questions have to be answered and the interview can stop at any time. Any data that could identify the individual practice, or patient, will be anonymised prior to any form of dissemination. Participants will have up to two weeks after the interview to withdraw their data prior to data analysis.

We will aim to conduct 30 interviews, however, interviews will be stopped when data saturation has been reached and no new major themes are arising within the last five transcripts.[59 60] Due to our interview topics being around a specific aspect of care, combined with interviewees having a similar role and training background we should achieve saturation within our recruitment target.[61] We will aim to include at least two interviews from each group of healthcare professionals (consultants; specialist registrars; GPs; GP registrars; practice managers; advanced nurse practitioners; community pharmacists). Purposive sampling will be used with snowballing to ensure that the correct mix of healthcare professionals is included.

## Data analysis

All interviews will be recorded using a digital voice recorder and consent for this will be obtained prior to starting. Digital recordings will then be transcribed verbatim prior to analysis.

The returned document will be checked, both for queries and to ensure that anonymity is protected prior to being imported into a suitable analysis programme such as NVivo (V.11).

TDF will be the main conceptual framework used in the qualitative work. This has been validated in a series of primary and secondary studies[29 31–36 62 63] as a method of understanding behaviour within a clinical setting. NPT will also be used and this framework has been developed to support the implementation and evaluation of complex interventions[38 64 65] and has been used extensively in trials, systematic reviews and qualitative work.[66] This theory will allow us to assess the 'key ingredients' when a guideline or local initiative has been successfully implemented in practice. This information will help us to include ingredients for successful implementation of our new tool to help manage patients with multimorbidity more effectively. Outside of these two frameworks, we will remain sensitive to new themes and will analyse iteratively.

## Study outcomes

The outcomes for the qualitative work are:
► To assess how guidelines and other local initiatives are being implemented in the management of patients with multimorbidity within primary and secondary care with the outcome of defining 'key ingredients' for successful implementation; and
► To understand the barriers and enablers around managing multimorbidityincluding which factors are taken into consideration in making clinical decisions.

## Phase II
### Site recruitment

Primary care centres who have agreed to take part in phase I and have EMIS as their clinical system will be approached for phase II.

### Participant recruitment/data collection

We will use custom-built code provided by PRIMIS (The University of Nottingham, Nottingham, UK, https://www.nottingham.ac.uk/primis/index.aspx) which generates a report within the GPs' clinical system. The code will be fully tested prior to use to ensure patient anonymity is protected. All eligible patients over the age of 18 years who have been registered at the practice for a minimum of three months will have data extracted relating to prescriptions. For patients to be fully eligible they must also have a Read Code within their notes at any time for at least one of the target conditions (supplementary file 1) and have had at least one prescription issued within the last two years. All data will remain anonymous and we will extract one possible identifier, sex, to allow us to assess the population demographic. To ensure anonymity, age at the date of extraction will also be collected rather than date of birth. The code will produce three reports; one will detail the practice profile, one will detail the

**Table 1** Estimates within UK primary care between 2014 and 2015

| | List size (total number) | List size (estimated 30–74-year-old patients) | CHD (number on register) | Hypertension (number on register) | COPD (number on register) | Diabetes mellitus, 17 years and above (number on register) | Depression (number on register) |
|---|---|---|---|---|---|---|---|
| Mean | 7304 | 4049 | 237 | 1007 | 133 | 375 | 425 |
| SD | 4429 | 2446 | 167 | 650 | 98 | 233 | 338 |
| IQR | 5870 | 3323 | 215 | 854 | 119 | 301 | 400 |

CHD, coronary heart disease; COPD, chronic obstructive pulmonary disease; SD standard deviation; IQR inter-quartile range.

conditions the patient has coded with the Read Code included and the final report will detail the drug history over the preceding 24 months.

## Sample size

This study uses a retrospective audit approach to collect data, therefore there is limited scope for increasing the sample size as there is not a traditional recruitment component. Using the Quality Outcomes Framework (QOF) data from 2014 to 2015, the average prevalence and list size has been calculated in table 1. Currently, there is no register of patients with osteoarthritis, so these numbers are not available. Table 2 shows the estimated numbers extrapolated from table 1, modelling the effects of practice number. It is likely that these are minimum estimates as patients may have associated Read Codes but have not been included on the disease register. Currently we have no data for the number of prescribed drugs that currently interact for this population. Assuming that 10% of drugs prescribed have an interaction, we can assume that a minimum of 53 and a maximum of 806 parameters can be assessed (table 2) which will be sufficient for the fixed-effects model. Therefore in this pilot study we aim to recruit a minimum of four primary care centres up to a maximum of eight with the overall intention of using these data to form the basis of a sample size calculation in a larger study.

## Data management

The QOF business rules V.34 will be used to extract Read Codes for five of our six target conditions (chronic obstructive pulmonary disease, type 2 diabetes, depression, coronary heart disease and hypertension). The general parent code will be used to extract patients suffering from osteoarthritis as there are currently no QOF indicators that are suitable for this (supplementary file 1). Every patient identified using these codes will have retrospective prescription data collected for the preceding 2 years. Unique subject identification numbers will be generated so that a comprehensive picture of prescriptions can be built up for each patient, while remaining anonymised. The clinical extraction code containing these details will be generated by a company with expertise in this area of work so that only relevant data are extracted and patient anonymity is preserved. We will not use a condition clustering approach and will only use the combinations of conditions as a way to characterise our population rather than drawing conclusions about prevalence of comorbidities.

After the patient's drug history has been constructed using a process mining approach, we will use our new drug interaction tool to assess the numbers and types of conflicts. This tool is based on the *British National Formulary* (*BNF*) interactions list, so we will use the same categories of no interaction, mild interaction and severe interaction. We will also use the expertise of ML to enable the use of natural language processing within this data set to pull out the types of interaction. In addition we will use any relating to drug clearance to reinterpret those patients with chronic kidney disease (CKD). This work will help inform the software based tool that is being developed within the grant. We will also extract data relating to CKD stage and we will use these data to interpret our findings due to changes in renal function often triggering a change in prescriptions. We recognise that there are challenges managing patients with multimorbidity, and that there may be a time when drugs that

**Table 2** Minimum and maximum estimates for selected conditions

| | 4 GP practices | 6 GP practices | 8 GP practices |
|---|---|---|---|
| Estimated average list size | 29 216 | 43 824 | 58 432 |
| Estimated average list of 30–74-year-old patients | 16 196 | 24 294 | 32 392 |
| Estimated number of patients with CHD | 948 | 1422 | 1896 |
| Estimated number of patients with hypertension | 4028 | 6042 | 8056 |
| Estimated number of patients with COPD | 532 | 798 | 1064 |
| Estimated number of patients with diabetes mellitus | 1500 | 2250 | 3000 |
| Estimated number of patients with depression | 1700 | 2550 | 3400 |

CHD, coronary heart disease; COPD, chronic obstructive pulmonary disease.

interact have to be used due to the clinical severity of one or more conditions. Therefore these data are not going to be linked to specific prescribers, and data will be fed back to practices in a pseudo-anonymised form.

A practice age demographic profile will be gained from the practice and publicly available QOF records, so that the results can be interpreted in the correct context. These results will also be compared with national records of disease profiling using the QOF disease registers and associated prevalence.

## Data analysis

Data will be analysed using a fixed-effects model with interaction terms. The interaction term will be whether a drug pair is appropriate or inappropriate as defined using the *BNF*, the current gold standard method,[67] with a variable of presence or absence of condition. Pairwise combinations of all conditions will be explored, as the literature suggests that each combination of two diseases has an evidence base. Our tool will complement existing prescribing support tools[68 69] as it places prescribing support within the context of navigating multiple care pathways.

## Study outcomes

The primary outcome for phase II is:

► To elicit the proportion of patients with pairwise condition combinations who have been prescribed two or more drugs concurrently, and the type of interaction, if any, between the two.

Further secondary outcomes for this work are to elicit:

► The total number, and severity, of interactions for each clinical condition as defined by the latest gold standard references.

► To compare the numbers of patients identified with these clinical conditions with the practice disease register and to the national reported prevalence using the latest available QOF data.

## DISCUSSION

This study protocol is using a mixed methods approach to better understand the challenges associated with multimorbidity and polypharmacy. In determining the utilisation of guidelines in the management of multiple morbidities we will use two methodological approaches. Both TDF[29 31–36 62 63] and NPT have been rigorously evaluated and used within the NHS context.[38 64–66] The dual analysis will enable a more robust understanding of the current issues. By extracting prescribing data on multimorbid patients for the first time we will be able to quantify the prevalence of polypharmacy in patients with combinations of some of the most common conditions. We will use this work to improve the implementability and acceptability of the new computer tool that is being formed as part of the MITCON grant. The MITCON tool will automatically detect conflict between guidelines to give healthcare professionals the most accurate and up-to-date information at the point of patient contact.

There are also a number of potential limitations with our approach. First, there is a risk of responder bias at both the practice recruitment and interviewee recruitment stages. Our findings will not necessarily be representative of the whole of the UK, however by conducting our study in the West Midlands we have the opportunity to recruit from a range of practices in terms of size and socioeconomic environment, and staff with a range of seniority and experience. Also, by conducting semi-structured interviews with individual members of staff we expect to reduce the impact of social desirability bias. Furthermore, the majority of our interviews in primary care are running the clinical system EMIS which may have different challenges to the other clinical systems currently running. We seek to address this by recruiting at a specialist educational meeting to include at least two GPs running a different clinical system. Also, during the training period, it is likely that GPs will have used other systems, so we will prompt around this area so that this limitation is minimised. Also, as this is a pilot study and we are not yet able to accurately estimate numbers of interactions we will focus on pairwise comparison initially. However, if it becomes apparent that many patients are taking three or four drugs, at the same time, that interact, we will analyse these separately. Furthermore, we are only able to assess six of the common morbidities in the UK, so there is a potential for missing excess morbidity as well as introducing bias relating to our chosen conditions. However, we have chosen prevalent chronic conditions which are all known to interact with each other, both within and between body systems. Finally, we are aware of larger studies assessing polypharmacy[70 71] but to the best of our knowledge we are the first study to undertake a mixed methods approach within each primary care site.

The findings of each phase will be important in their own right but taken together they will provide a rigorous and robust description of the current issues around managing multimorbidity and the prevalence of polypharmacy. This information will prove invaluable in the development of our novel software tool to ensure that it can be tailored to barriers most amenable to change.

## TRIAL STATUS

At the time of submission, this study was actively recruiting centres and participants and no analysis work has yet been undertaken.

**Contributors** RB wrote the first draft of the protocol and all authors participated in further drafting. AMT provided specialist clinical knowledge. PW and ML provided specialist IT knowledge for the analysis of the polypharmacy data. RB and IL conceived the original idea for this study. All authors read and approved the final manuscript.

**Funding** This work was supported by the Engineering and Physical Sciences Research Council (Grant No. EP/M014401/1). The custom-built code was funded by the University of Birmingham College of Medical and Dental Science Research Development Fund.

**Competing interests** None declared.

**Ethics approval** The University of Birmingham Research Ethics Committee.

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
