## [Reviewer comments · BMJ Open]

ARTICLE DETAILS

TITLE (PROVISIONAL)	Assessing the extent of drug interactions amongst patients with multimorbidity in primary and secondary care in the West Midlands (UK): A study protocol for the Mixed Methods Multimorbidity Study (MiMMS)
AUTHORS	Backman, Ruth; Weber, Philip; Turner, Alice; Lee, Mark; Litchfield, Ian.

VERSION 1 - REVIEW

REVIEWER	Susan Smith RCSI Dublin, Ireland
REVIEW RETURNED	13-Apr-2017

GENERAL COMMENTS	• The title needs to be more specific to let readers know what the study is actually about beyond just stating 'assessing multimorbidity'• The background predominantly focuses on CDSS. The novelty of this study is the stated aim to develop a computer tool to support multimorbidity management. If this has a prescribing focus, as suggested by the phase 2 study, there needs to be more consideration of existing deprescribing support tools and interventions to address potentially inappropriate prescribing using computerized support systems (see deprescribing.org and https://www.ncbi.nlm.nih.gov/pubmed/27321600 for examples).• Where does this proposed intervention fit in to the recent NICE Guidance on multimorbidity• There is already qualitative literature examining multimorbidity management. The authors quote some of these papers but do not seem to be building on existing findings. What is novel about this study is the primary and secondary care focus simultaneously and this could be explored more.• However, it is unclear what type of secondary care services will be included – challenge in that these may be condition focused unless can interact with Medicine for the Elderly, which then leaves out the middle aged multimorbidity group. Perhaps the secondary care focus is condition specific but asking specialists how they manage multimorbidity in patients in their diabetes clinics etc. do they even see it as their role?• My understanding and previous use of NPT suggests it is used to consider a specific intervention or service and it is not clear how this applies in this study given how broad the objectives are. Are the authors seeking to use NPT to explore whether a computer tool type intervention for MM would be feasible and acceptable and could be 'normalised' into practice?• The authors state that they may do a comparative analysis between the frameworks if time allows – this needs to be decided at protocol
--

	stage and should either be included or removed from the protocol as an aim  • End of description of phase 1, study objectives seem to be described as study outcomes. Again these are very broad objectives and would be better to be more specific as to what this study will add to existing evidence • Phase 2: Observational study. This study aims to determine the extent of polypharmacy in the West Midlands based on prescribing data from 4 practices. There are already papers presenting this using national datasets in the UK. – see work of Guthrie et al. This needs to be referred to and a justification as to why this smaller study will add to the literature • It is also unclear how this study of polypharmacy will support the development of the on-line tool, which is presumably its overall purpose and is what makes it more novel • The authors state that sample size is limited by the number of patients in these practices. This is reasonable but the existing studies examining condition clustering use far higher numbers of patients, which I understand is needed for cluster type analyses. There is also existing work on condition clustering, see https://academic.oup.com/fampra/article/32/5/505/689101/Multimorbidity-patterns-in-a-primary-care and amongst other papers) • Page 15, lines 16,17. This observational study is described here as a feasibility study – what is it testing the feasibility of? This term is usually used for studies in which interventions are being developed or pilot tested. If this is the intention with this study, it is not clear from how it is reported • Page 15, line 13: what is meant by an interaction? Need to be clearer in definitions – is it to do with potential for adverse drug events or PIP? • There are some inconsistencies throughout as to whether there will be 4 or 8 primary care centres – it's quite a distinction • Page 12, line 13 – I'm not familiar with PRIMIS, please use full name and provide reference • Table 2: many of the condition types will overlap and not clear how this will be handled – is the analysis being undertaken by condition or by patient? • Once drug histories for each patient are constructed, the authors plan to use what they describe as a new drug interaction tool – is this the tool they are developing overall. There is no reference provided for this and no further detail? • Outcomes for phase 2 seem strange. Most people with two conditions will be prescribed at least 2 medicines. The drug interaction focus is very unclear given the lack of a definition and need for clarity around potential seriousness and health impact of interaction – why not use one of the widely used PIP criteria like Beers or Stopp Start. Again authors mention a gold standard here but no reference • Discussion: Would be useful for authors to consider what aspect of multimorbidity they are targeting. See http://health.gov.ie/wp-content/uploads/2016/12/Prof-Bruce-Guthrie-Presentation.pdf
--	--

REVIEWER	Juliane Koeberlein-Neu University of Wuppertal, Germany
REVIEW RETURNED	26-Apr-2017

GENERAL COMMENTS	The protocol is well written and method is presented in a transparent way. I have only a few minor remarks:
---

	1) I missed a clear definition of multimorbidity and polypharmacy, which will be applied in this study. 2) A figure, which displays the study method - esp. the recruitment process - would enhance the comprehensibility of this paper. 3) In my opinion, Tab 1 (Read codes) could be placed in the supplement file. 4) Studies from other countries, addressing the same research question, should be cited in the introduction part of this paper. 5) There seems to be a citation failure in line 41-46, page 5. Reference is displayed in a wrong format.
--	---

REVIEWER	Ingmar Schäfer Department of Primary Care, University Medical Center Hamburg-Eppendorf, Germany
REVIEW RETURNED	10-May-2017

GENERAL COMMENTS	pg. 5, ln 13: "Polypharmacy is an inevitable part of following many sets of guidance ..." This is not necessarily so! But the likelihood of polypharmacy may be increased if multiple sets of guidelines are used for one patient and all recommendations from these guidelines are fully applied. pg. 7, ln. 20: "... purposely sample up to four secondary care centres with differing patient demographics..." & pg. 10 ln. 43: "Purposive sampling will be used ..." I would be interested in the rationale choosing a sample of four centers (and two pharmacies). There is also some information needed about which demographic data are used for purposeful sampling and which cut off values are used, if applicable (e.g. mean age 18-50 vs. 50+). pg. 9 ln. 13: "... until data saturation has been achieved." Please give a definition of data saturation, e.g. no new categories in the last three interviews. pg. 10 ln. 34: "... a maximum of 30 interviews." Please give the rationale for this maximum sample size and discuss what happens if data saturation has not been reached in interview no. 30. Possibly, this needs to be listed as a study limitation. pg. 14 ln. 48: "This study uses an audit approach to collect data, therefore there is limited scope for increasing the sample size..." I do not agree with this statement. There might be so many cases in the data sets that the needed sample size is easily reached, but a protocol needs to state how many patients are needed for the analyses (and a rationale for this) and also, how many cases can be expected from the data extraction. This information needs to be included in the text. pg. 17, ln. 27: "... the current gold standard method..." This (or these) method(s) need(s) to be introduced and referenced. pg. 17, ln. 29: "Pairwise combinations of all conditions will be explored..." pg. 17, ln. 43: "... patients with pairwise combinations who have been described two or more drugs..." In my experience
--

	there are very few patients with exactly two diseases. What about the additional diseases in each person? Are people with more than two (and less than two) conditions excluded from the study? Is "excess morbidity ignored? And what about conditions not assessed in the study, e.g. cancer? This needs to be detailed. And if only two-way interactions are considered, what about three-way, four-way, etc. interactions? This is a study limitation! I would also be interested in a complete list of variables used in study phase two. This is needed to ensure replication of the study.
--	---

VERSION 1 – AUTHOR RESPONSE

Reviewer 1

The title needs to be more specific to let readers know what the study is actually about beyond just stating 'assessing multimorbidity'

- Thank you for your comment, this has now been adjusted within the manuscript.

Where does this proposed intervention fit in to the recent NICE Guidance on multimorbidity

- The ultimate aim of the overriding MITCON grant is to produce a computer based tool that can be used by healthcare professionals. This tool will automatically detect conflicts between all NICE guidelines, including recommendations from the multimorbidity guidance. This will enable clinicians to have the latest evidence based information, including conflicts of medication and advice, to make a clinical decision based on the patient's preferences and values, which is one of the recommendations within the multimorbidity guidelines. If successful, this tool would be tested before being introduced into practice.

The background predominantly focuses on CDSS. The novelty of this study is the stated aim to develop a computer tool to support multimorbidity management. If this has a prescribing focus, as suggested by the phase 2 study, there needs to be more consideration of existing deprescribing support tools and interventions to address potentially inappropriate prescribing using computerized support systems (see deprescribing.org and <https://www.ncbi.nlm.nih.gov/pubmed/27321600> for examples).

- Thank you for your comment. Prescribing is not the main focus of this project, but will need to be a consideration within the tool. We will use the information we collect within the study alongside the evidence when forming the tool to ensure that it is relevant and up to date.

There is already qualitative literature examining multimorbidity management. The authors quote some of these papers but do not seem to be building on existing findings. What is novel about this study is the primary and secondary care focus simultaneously and this could be explored more.

- Thank you for your comment. We have added some text to highlight the novelty and benefit of assessing barriers in multiple healthcare contexts and how we are using a mixed methods approach to relate the data in specific GP practices.

However, it is unclear what type of secondary care services will be included – challenge in that these may be condition focused unless can interact with Medicine for the Elderly, which then leaves out the middle aged multimorbidity group. Perhaps the secondary care focus is condition specific but asking specialists how they manage multimorbidity in patients in their diabetes clinics etc. do they even see it as their role?

- Thank you for your comment. We have added a line in the text to confirm that we will target consultants who practice in a speciality relevant to our target conditions as well as elderly care physicians if possible.

From preliminary data we are able to confirm that some consultants do see managing multimorbidity as part of their role as some reported that they felt they had a stronger relationship with the patient than any other member of their healthcare team, so we hope to be able to add this information to the

evidence base in due course.

My understanding and previous use of NPT suggests it is used to consider a specific intervention or service and it is not clear how this applies in this study given how broad the objectives are. Are the authors seeking to use NPT to explore whether a computer tool type intervention for MM would be feasible and acceptable and could be 'normalised' into practice?

- In this work we are trying to identify factors where computer based tools and guidance have been successfully implemented. We would then like to ensure that our tool has as many of these 'key ingredients' as possible to help the tool be the most implementable alongside the many challenges within the context.

The authors state that they may do a comparative analysis between the frameworks if time allows – this needs to be decided at protocol stage and should either be included or removed from the protocol as an aim

- Thank you for your comment, we have decided to remove this from the protocol manuscript.

End of description of phase 1, study objectives seem to be described as study outcomes. Again these are very broad objectives and would be better to be more specific as to what this study will add to existing evidence.

- Thank you for your comment, we have altered the text to refine the outcomes.

Phase 2: Observational study. This study aims to determine the extent of polypharmacy in the West Midlands based on prescribing data from 4 practices. There are already papers presenting this using national datasets in the UK. – see work of Guthrie et al. This needs to be referred to and a justification as to why this smaller study will add to the literature

- We appreciate that this is a small scale study, however we hope that this will add to the literature as it is the first time that GPs have been interviewed about their perspective on polypharmacy and then looked at their own retrospective prescribing data and have added this to the limitations section.

It is also unclear how this study of polypharmacy will support the development of the on-line tool, which is presumably its overall purpose and is what makes it more novel

- Currently the information relating to polypharmacy will not be directly integrated within this tool and it forms a future direction for our group. Should there be any relevant findings around numbers of over the counter medications being taken, or items that are considered high risk prescribing, we will try to include relevant prompts to this in our tool.

The authors state that sample size is limited by the number of patients in these practices. This is reasonable but the existing studies examining condition clustering use far higher numbers of patients, which I understand is needed for cluster type analyses. There is also existing work on condition clustering, see <https://academic.oup.com/fampra/article/32/5/505/689101/Multimorbidity-patterns-in-a-primary-care-and-amongst-other-papers>)

- Thank you for your comment and the inclusion of this paper. We fully agree that our study would not be able to assess condition clustering and we will include the numbers of patients with the combinations of conditions as a way to characterise our population rather than drawing conclusions about prevalence of comorbidities in the final manuscript.

Page 15, lines 16,17. This observational study is described here as a feasibility study – what is it testing the feasibility of? This term is usually used for studies in which interventions are being developed or pilot tested. If this is the intention with this study, it is not clear from how it is reported

- Thank you for your comment. We have altered this to read 'pilot' as we are not testing an intervention. We are aware that this is a small study but we are planning a larger scale version for the future and we will use this work to help guide the larger study.

Page 15, line 13: what is meant by an interaction? Need to be clearer in definitions – is it to do with potential for adverse drug events or PIP?

- Thank you for your comment. This wording has been altered to add in our study definition of interaction. We will initially class these as 1) no drug interaction, 2) a mild drug interaction that would cause no consequences to the patient, and 3) a severe interaction. We will also use the expertise of a team member so that Natural Language Processing can be used within this data set to pull out the types of interaction, and will use any relating to drug clearance to reinterpret those patients with chronic kidney disease.

There are some inconsistencies throughout as to whether there will be 4 or 8 primary care centres – it's quite a distinction

- Thank you for this comment, the manuscript has been checked and edited to read a minimum of four to a maximum of eight primary care centres will be recruited

Page 12, line 13 – I'm not familiar with PRIMIS, please use full name and provide reference

- Thank you for your comment, this has been addressed in the text.

Table 2: many of the condition types will overlap and not clear how this will be handled – is the analysis being undertaken by condition or by patient?

- The analysis will be undertaken by patient rather than condition. We will then use the conditions the patient has as the context to interpret the results.

Once drug histories for each patient are constructed, the authors plan to use what they describe as a new drug interaction tool – is this the tool they are developing overall. There is no reference provided for this and no further detail?

- We plan to use a process mining approach to analyse the drug histories and have a prototype tool for this which has not yet been written up. We have clarified this within the text.

Outcomes for phase 2 seem strange. Most people with two conditions will be prescribed at least 2 medicines. The drug interaction focus is very unclear given the lack of a definition and need for clarity around potential seriousness and health impact of interaction – why not use one of the widely used PIP criteria like Beers or Stopp Start. Again authors mention a gold standard here but no reference

- This wording has been slightly altered and hopefully with the definition now in place this will be stronger as an outcome. In this work we are not looking to assess high risk prescribing or inappropriate prescribing, we are just looking at the scope of polypharmacy in the West Midlands.

Discussion: Would be useful for authors to consider what aspect of multimorbidity they are targeting. See <http://health.gov.ie/wp-content/uploads/2016/12/Prof-Bruce-Guthrie-Presentation.pdf>

- Thank you for your comment and sending this useful presentation. We have added some text to the discussion to show how this work will relate to the larger MITCON grant by making a tool that has had initial engagement from key stakeholders to help improve its usability and implementability in the longer term.

Reviewer 2

I missed a clear definition of multimorbidity and polypharmacy, which will be applied in this study.

- Thank you for your comment, this has been added to the research questions at the bottom of the

introduction.

A figure, which displays the study method - esp. the recruitment process - would enhance the comprehensibility of this paper.

- Thank you for your comment, this has now been added to the manuscript as Figure 1.

In my opinion, Tab 1 (Read codes) could be placed in the supplement file.

- Thank you, this table has been moved to supplementary file 1.

Studies from other countries, addressing the same research question, should be cited in the introduction part of this paper.

- Thank you for your comment. We are very happy to add these to the introduction and would be grateful if you could provide the specific references to allow us to address this comment fully.

There seems to be a citation failure in line 41-46, page 5. Reference is displayed in a wrong format.

- Thank you for your comment. This is citing a currently unpublished protocol for the MITCON programme, so we have followed the Journals' rule regarding this type of citation.

Reviewer 3

pg. 5, In 13: "Polypharmacy is an inevitable part of following many sets of guidance ..." This is not necessarily so! But the likelihood of polypharmacy may be increased if multiple sets of guidelines are used for one patient and all recommendations from these guidelines are fully applied.

- Thank you for your comment. This wording has been altered within the manuscript.

pg. 7, In. 20: "... purposely sample up to four secondary care centres with differing patient demographics..." & pg. 10 In. 43: "Purposive sampling will be used ..." I would be interested in the rationale choosing a sample of four centers (and two pharmacies). There is also some information needed about which demographic data are used for purposeful sampling and which cut off values are used, if applicable (e.g. mean age 18-50 vs. 50+).

- Due to this study having no direct payment associated with recruitment, we recruited all sites who took an interest and were not able to formally balance demographics. We did recruit from both the north and south side of the city and will publish the demographics of the recruited sites in the final manuscript.

pg. 9 In. 13: "... until data saturation has been achieved." Please give a definition of data saturation, e.g. no new categories in the last three interviews.

- Thank you for your comment, this has been fully defined within the text.

pg. 10 In. 34: "... a maximum of 30 interviews." Please give the rationale for this maximum sample size and discuss what happens if data saturation has not been reached in interview no. 30. Possibly, this needs to be listed as a study limitation.

- Thank you for your comment. We recognise the limitations of specifying a number in qualitative research as all interviewees will have a different personal experience they wish to share. However, in many similar studies where barriers and enablers are assessed within a healthcare setting, between 20 and 30 interviews are usually specified. We will add this as a limitation to the study findings in the subsequent manuscript.

pg. 14 In. 48: "This study uses an audit approach to collect data, therefore there is limited scope for increasing the sample size..." I do not agree with this statement. There might be so many cases in the

data sets that the needed sample size is easily reached, but a protocol needs to state how many patients are needed for the analyses (and a rationale for this) and also, how many cases can be expected from the data extraction. This information needs to be included in the text.

- This wording has been altered to read 'retrospective audit'. Although we are able to estimate the numbers of patients with single conditions, we were not able to identify the numbers of patients within our population who had a combination of at least two of these conditions. Furthermore, we were not able to estimate the number of drug interactions with this group, so we have based our calculations on 10% of the population having an interaction. We were advised by our statistician that recruiting a minimum of four practices of average size would allow us to perform basic pairwise comparison tests so that we could use this feasibility study to perform a full sample size calculation for a larger trial.

pg. 17, ln. 27: "... the current gold standard method..." This (or these) method(s) need(s) to be introduced and referenced

- Thank you for your comment. We are classifying interactions as defined by the BNF and this has now been referenced within the text.

pg. 17, ln. 29: "Pairwise combinations of all conditions will be explored..." pg. 17, ln. 43: "... patients with pairwise combinations who have been described two or more drugs..." In my experience there are very few patients with exactly two diseases. What about the additional diseases in each person? Are people with more than two (and less than two) conditions excluded from the study? Is "excess morbidity ignored? And what about conditions not assessed in the study, e.g. cancer? This needs to be detailed. And if only two-way interactions are considered, what about three-way, four-way, etc. interactions? This is a study limitation!

VERSION 2 – REVIEW

REVIEWER	Susan Smith RCSI, Ireland
REVIEW RETURNED	19-Jun-2017

GENERAL COMMENTS	The authors have addressed my previous comments
---

REVIEWER	Ingmar Schäfer University Medical Center Hamburg-Eppendorf, Germany
REVIEW RETURNED	19-Jun-2017

GENERAL COMMENTS	1) I still think that limiting the number of cases to 30 is a serious study limitation that needs to be mentioned in the manuscript I assessed and not only in other publications. 2) In your reply to my comment no. 5 you write: "We were advised by our statistician that recruiting a minimum of four practices of average size would allow us to perform basic pairwise comparison tests so that we could use this feasibility study to perform a full sample size calculation for a larger trial." In my opinion, this needs to be discussed in the manuscript. 3) You reply to my comment vo. 7: "We agree with your thoughts around diagnosis, particularly around symptoms and conditions. [...]"
---

	We recognise that this is a limitation of the work and this will be highlighted in all future publications." I would also expect this problem to be discussed in the manuscript I reviewed.
--	---

VERSION 2 – AUTHOR RESPONSE

Reviewer 3

1) I still think that limiting the number of cases to 30 is a serious study limitation that needs to be mentioned in the manuscript I assessed and not only in other publications.

Further to our previous reply, we have performed an outline scoping exercise looking for similar studies where healthcare professionals have been interviewed to assess barriers and enablers using the Theoretical Domains Framework. Within the first ten relevant references we found that seven interviewed up to 30 people, and one further study conducted 20 semi structured interviews and supplemented these with focus groups. We appreciate that the more healthcare professionals you interview, the more sure you can be of reaching data saturation. However, we feel that interviewing up to 30 is reasonable, so we have altered the manuscript to remove the word 'maximum' so that if more interviews are required, we will continue until data saturation has been reached (page 10, last paragraph).

Cameron J Phillips Experiences of using the Theoretical Domains Framework across diverse clinical environments: a qualitative study (2015) No. interviewed 10

Neil Roberts What helps or hinders the transformation from a major tertiary center to a major trauma center? Identifying barriers and enablers using the Theoretical Domains Framework (2016) No. interviewed 13

Brian T Power Understanding perceived determinants of nurses' eating and physical activity behaviour: a theory-informed qualitative interview study (2017) No. interviewed 16

Deborah Debono Applying the Theoretical Domains Framework to identify barriers and targeted interventions to enhance nurses' use of electronic medication management systems in two Australian hospitals (2017) No. interviewed 19

Anna Chapman Barriers and enablers to the delivery of psychological care in the management of patients with type 2 diabetes mellitus in China: a qualitative study using the theoretical domains framework (2016) No. interviewed 23 (2 x focus groups)

Annie McCluskey Barriers and enablers to implementing multiple stroke guideline recommendations: a qualitative study (2013) No. interviewed 28

Kerry Murphy Understanding diagnosis and management of dementia and guideline implementation in general practice: a qualitative study using the theoretical domains framework (2014) No. interviewed 30

Oyun Chimeddamba Implementation of clinical guidelines on diabetes and hypertension in urban Mongolia: a qualitative study of primary care providers' perspectives and experiences (2015) no. interviewed 20 (in focus groups), 20 (semi structured interviews)

Emily Bain Barriers and enablers to implementing antenatal magnesium sulphate for fetal neuroprotection guidelines: a study using the theoretical domains framework (2015) no. interviewed 45

Rebecca Lawton Using the Theoretical Domains Framework (TDF) to understand adherence to multiple evidence-based indicators in primary care: a qualitative study (2016) no. interviewed 60

2) In your reply to my comment no. 5 you write: "We were advised by our statistician that recruiting a minimum of four practices of average size would allow us to perform basic pairwise comparison tests so that we could use this feasibility study to perform a full sample size calculation for a larger trial." In my opinion, this needs to be discussed in the manuscript.

Thank you for your comment. We have added the following to the manuscript 'Therefore in this pilot study we aim to recruit a minimum of four primary care centres up to a maximum of eight with the overall intention of using this data to form the basis of a sample size calculation in a larger study.'

3) You reply to my comment vo. 7: "We agree with your thoughts around diagnosis, particularly around symptoms and conditions. [...] We recognise that this is a limitation of the work and this will be highlighted in all future publications." I would also expect this problem to be discussed in the manuscript I reviewed.

Thank you for your comment; we have added this as a limitation to the current manuscript alongside the limitations of a pairwise approach. "Furthermore, we are only able to assess six of the common morbidities in the UK, so there is a potential for missing excess morbidity as well as introducing bias relating to our chosen conditions. However, we have chosen prevalent chronic conditions which are all known to interact with each other, both within and between body systems." (Page 18, paragraph 1

VERSION 3 – REVIEW

REVIEWER	Ingmar Schäfer University Medical Center Hamburg-Eppendorf, Germany
REVIEW RETURNED	14-Jul-2017

GENERAL COMMENTS	All of my concern have been addressed properly.
---